# Association of 100% Fruit Juice Consumption with Cognitive Measures, Anxiety, and Depression in US Adults

**DOI:** 10.3390/nu14224827

**Published:** 2022-11-15

**Authors:** Sanjiv Agarwal, Victor L. Fulgoni, Paul F. Jacques

**Affiliations:** 1NutriScience LLC, East Norriton, PA 19403, USA; 2Nutrition Impact LLC, Battle Creek, MI 49014, USA; 3Jean Mayer USDA Human Nutrition Research Center on Aging, Tufts University, Boston, MA 02111, USA

**Keywords:** national health and nutrition examination survey, NHANES, neurocognitive markers

## Abstract

Cognitive decline, anxiety, and depression are significant contributors to human ill health and decreased quality of life. Intake of fruits including 100% juice is associated with beneficial effects on several health outcomes. The current study explored the potential associations between consumption of 100% fruit juice and neurocognitive markers in adults. Twenty-four-hour dietary recall data from the National Health and Nutrition Examination Survey (NHANES) 1988–1994, 1999–2018 for adults 20+ years (N = 62,606) were used to assess 100% fruit juice intake, and NHANES questionnaire data were used for the assessment of neurocognition. Association of usual intake of 100% fruit juice with neurocognitive outcomes were assessed by regression analysis after adjusting for demographic, lifestyle, dietary, and health-related covariates. About 21% of adults were fruit juice consumers and the intake was a little over 2 oz per day. Consumers of 100% fruit juice had 16.3% fewer days per month of feeling anxious compared to non-consumers. There were no significant associations of 100% fruit juice usual intake with other neurocognitive measures. A large number of potential confounding variables have been identified to be associated with one or more neurocognitive measures suggesting that these may be important parameters to include in future studies.

## 1. Introduction

Neurocognitive functions are closely linked to the function of neural pathways, or cortical networks in the brain. Cognitive decline is a major cause of disability in old age [1,2,3] and is a major public health challenge with increased associated health care cost [4,5,6]. Depression and mental distress are also a very significant contributor to human ill health and decreased quality of life [7,8,9]. Healthy lifestyle, including exercise and healthy diet, are consistently recommended to improve memory performance and brain function. Diet and nutrition are considered as modifiable determinants of neurocognitive health including mental stress, mood, memory function, cognitive decline, and development of dementia including Alzheimer’s disease [10,11,12,13,14]. The Mediterranean dietary pattern has been associated with lower risk of cognitive impairment, dementia, and Alzheimer’s disease [15,16,17]. Plant-based dietary patterns, including fruits and vegetables, have been linked with beneficial effects on cognition and reduced risk of cognitive decline and mental distress [18,19,20,21].

Increased consumption of fruit and vegetables has been consistently recommended by Dietary Guidelines worldwide. Dietary Guidelines for Americans 2020–2025 (DGA) also recommend intake of fruits and vegetables as part of a healthy dietary pattern [22]. Fruit and vegetable-rich diets have been reported to be associated with a reduced risk of chronic diseases and related mortalities [23,24,25]. However, almost 80–90% of Americans do not meet the fruit and vegetable intake recommendations [22], and intakes of fresh, frozen, or dried whole fruit, or 100% fruit juice are all counted toward meeting the fruit requirements. One cup of 100% fruit juice is considered as a one cup serving from the “Fruit Group”, and the DGA indicate that 100% fruit juice can provide that up to 50% of the recommended amount of fruit in a healthy dietary pattern [22,26]. In a recent dietary modeling study, we reported that, except for a small increase in dietary fiber, there was no adverse impact on nutrient intake when whole fruits were replaced by 100% fruit juice [27]. Intake of 100% fruit juices has been shown to be associated with an increased intake of nutrients and improved diet quality in cross-sectional studies [27,28,29]. There is some research that suggests that 100% juice intake is associated with beneficial effects on body weight and other health outcomes; however, the data are largely inconclusive [30,31,32,33,34,35]. An analysis of data from the 2017 Behavioral Risk Factor Surveillance System found that consumption of 100% fruit juice was not associated with experiencing 14 or more days of poor mental health in the past month, but these authors suggested future studies should evaluate if certain larger amounts of fruit juice consumption were associated with mental health [36].

The National Health and Nutrition Examination Survey (NHANES) is a very large, nationally representative, and currently a continuous cross-sectional survey of noninstitutionalized Americans that is designed to monitor the health and nutritional status of the population [37]. The survey examines a nationally representative sample of about 10,000 adults and children every two years and includes an in-home survey and a mobile laboratory physical examination with detailed questionnaires to assess health, including mental health. The NHANES survey also assesses food consumption and dietary supplement use, and health-related parameters of participants using demographic, socioeconomic, dietary, and health-related questionnaires. Standardized, state-of-the-art procedures are used in conducting the NHANES surveys, and therefore very large datasets can be obtained by combining multiple years of data. Data from these surveys are used to assess nutritional status and its association with health promotion and disease prevention and assist with formulation of national standards and public health policy [37].

The current study was designed to explore the potential association between usual intake of 100% fruit juice and neurocognitive markers in adults using the NHANES dataset. We hypothesize that since 100% fruit juices have comparable nutrient profiles as fruits and their intake is associated with improved diet quality [27], their consumption would be beneficially associated with markers of mental health.

## 2. Materials and Methods

### 2.1. Database

Data from eleven NHANES cycles (1988–1994, 1999–2000, 2001–2002, 2003–2004, 2005–2006, 2007–2008, 2009–2010, 2011–2012, 2013–2014, 2015–2016 and 2017–2018) were used for the analysis. The NHANES data are collected via an in-home interview for demographic and basic health information and a comprehensive diet and health examination in a mobile examination center using a complex stratified multistage cluster sampling probability design. Details of the subject recruitment, survey design, and data collection procedures are available online [37], and all data obtained from this study are publicly available at: http://www.cdc.gov/nchs/nhanes/ (accessed on 16 October 2022). The National Center of Health Statistics Ethics Review Board approved the NHANES protocol, and signed written informed consent was collected from all participants or proxies. This study did not require Institutional Review Board review, as it was a secondary data analysis that lacked personal identifiers.

### 2.2. Study Population

We used the data from free living adults age 20+ years participating in NHANES 1988–1994 and NHANES 1999–2018 (n = 71,060), after excluding pregnant or lactating females (n = 2107), those with incomplete dietary data (n = 6341), and those with no calorie intake on day 1 (n = 6). The final sample size was 62,606 adults.

### 2.3. Estimation of Dietary Intake

In-person 24 h dietary recall interviews that were administered using an automated, multiple-pass (AMPM) method were used to obtain dietary intake data [38]. Moreover, 100% juice was defined by food codes in the What We Eat in America (WWEIA) subgroup 70 “100% Juice”. Those consuming any amount of 100% fruit juice during the first 24 h recall were defined as fruit juice consumers and were further classified based on specific levels of 100% fruit juice consumption: >0 to 4 oz, >4 to 8 oz, >8 to 12 oz, and >12 oz.

### 2.4. Estimation of Neurocognitive Markers

NHANES questionnaire data for different neurocognitive markers were used for the assessment of neurocognition in adult 100% fruit juice consumers and non-consumers. The following neurocognition markers were evaluated:Neurobehavioral Evaluation System 2 (NES2) consisting of the simple reaction time task (SRTT) measuring visuomotor speed, the symbol digit substitution test (SDST) measuring information-processing speed, and the single digit learning test (SDLT) measuring learning and recall were used to measure cognitive function.Data on the Consortium to Establish a Registry for Alzheimer’s Disease (CERAD) Word List Learning Test, the CERAD Word List Recall Test, the Animal Fluency test (AFT), and the Digit Symbol Substitution Test (DSST) were used to measure cognitive function.Response to the question “During the past 7 days, how often have you had trouble remembering where you put things like keys or wallet?” was used to assess memory impairment and its severity as a measure of dementia.Response to the question “How often do you feel worried, anxious?” and “In the past 12 months, did you have a period of a month or more when most days you felt worried or tense or anxious about everyday problems such as work or family?” were used to measure anxiety.Response to the question “Over the last 2 weeks, how often have you been bothered by the following problems: feeling down, depressed, or hopeless?” was used to measure depression.

Data for different neurocognitive markers were not collected in each NHANES cycle, data for some markers were collected in some cycles, and the data for other markers were collected in other cycles (Table 1). Therefore, the analyses for specific neurocognitive markers were limited to the NHANES cycles and age groups in which the markers were available.

### 2.5. Statistics

SAS 9.4 (SAS Institute, Cary, NC, USA) software was used for all analyses. Appropriate survey weights, strata, and primary sampling units were used to adjust the data for the complex sampling design of NHANES, and day 1 dietary weights were used in all analysis. Usual intakes of 100% fruit juice were determined by the National Cancer Institute method [39]. Least square means (LSM) and standard errors (SE) were generated via regression analyses for neurocognitive outcomes in non-consumers and 100% fruit juice consumers (including consumers by consumption level). Age, gender, ethnicity, physical activity level, poverty income ratio level, weight status, and current smoking status were used as covariates to adjust the data (Model 1). The Healthy Eating Index 2015 score was added as an additional covariate in Model 2; and weekday hours of sleep, antidepressant medication use, hypertension medication use, education level, glycohemoglobin, elevated blood pressure, whether a doctor told you that you had a stroke, and intakes of caffeine, vitamin B12, iron, folic acid, total fruits, whole fruits, total vegetables excluding legumes were additional covariates in Model 3. In linear trend analysis, the juice intake amount was added to the model as a continuous variable, while the juice consumption group numbers (juice consumption levels) were added to the model as a variable in group trend analysis. A *p*-value of <0.05 was used for statistical significance.

## 3. Results

Approximately 20.8, 18.4, and 27.6% of adults age 20+, 20–59, and 60+ years, respectively, were 100% fruit juice consumers and consumed 71.2, 70.3, and 73.8 g per capita (2.4, 2.3, and 2.5 oz per capita) of 100% juice on day 1 of dietary recall. Usual intakes of 100% fruit juice were 69.1, 67.7, and 72.9 g/d (2.22, 2.18, and 2.34 oz/d) for adults age 20+, 20–59, and 60+ years, respectively (Table 2).

Table 3 shows the differences in neurocognitive outcomes among 100% fruit juice consumers and non-consumers. Consumers of 100% fruit juice had 1.00 (16.3%) and 1.26 (18.6%) fewer days per month of feeling anxious compared to non-consumers among adults age 20+ years and 20–59 years, respectively, after adjusting the data in Model 1 and the differences remained significant after adjusting the data for additional covariates in Model 2 and Model 3. Usual intake of 100% fruit juice was inversely associated with days of feeling anxious among adults age 20+ and 20–59 years. However, the inverse association remained significant in Model 2 only in adults 20+ years and was attenuated in adults age 20–59 years and in Model 3 in both age groups. In adults age 60+ years, 100% fruit juice consumption was inversely associated with digit symbol scores in all three models. Results assessing association of 100% fruit juice consumption with all other markers were not significant.

Regarding analyses of specific levels of juice consumption on neurocognitive markers (Table 4), there were fewer days of feeling anxious in adults 20+ and 20–59 years consuming 4–8 ounces of 100% fruit juice in Model 1 and Model 2, and among adults 20+ consuming 8–12 ounces of 100% fruit juice in Model 1. There was also a significant inverse association with 100% fruit juice consumption levels and feeling anxious among adults 20+ and 20–59 years in group trend analysis in Model 1 and 2. Among adults age 60+ years, there was a 10.7% lower digital symbol score with >12 ounce of 100% fruit juice consumption in Models 1, 2 and 3. There were also an inverse association for the digital symbol score in Model 3 and a positive association for the trouble remembering in Model 1 with 100% fruit juice consumption levels among adults age 60+ years. Results for all other markers were not significant.

In Model 3, we assessed the relationship of a number of possible variables that may influence the relationship of fruit juice intake with neurocognitive measures (Appendix A). While some variables were associated (positively or inversely) with only one or two neurocognitive measures (e.g., total HEI score, vitamin B12 intake, and iron intake), several variables were associated with six or more of the neurocognitive measures (e.g., doctor told had a stroke, total vegetable intake, weekday sleep hours, anti-depression medication use, and education level).

## 4. Discussion

Results of the present analysis of NHANES (1988–1994, and 1999–2018) covering over three decades indicates that 20–30% of US adults consume 100% fruit juice and that intake was a little over 2 oz per day on a per capita basis. The regression analysis indicated that after adjusting the data for various demographic, lifestyle, dietary, and health-related factors, 100% juice intake was not associated with many of the neurocognitive measures but was inversely associated with frequency of anxiety among adults and with lower digital symbol scores in those 60+ years.

Although anxiety is a normal reaction to stress, anxiety disorders are the most common of mental disorders affecting over 20% of adults each year and nearly 30% of adults at some point in their lives. There are several types of anxiety disorders including panic disorder, phobias, social anxiety disorder, separation anxiety disorder, obsessive–compulsive disorder and generalized anxiety disorder. Anxiety disorder can affect daily activities such as job performance, schoolwork, and relationships [40,41,42]. Anxiety disorders can also affect cardiovascular, digestive, respiratory, and immune systems and can cause headaches, muscle tension, insomnia, depression, and social isolation [43]. The present regression analysis of cross-sectional data from NHANES shows that the adults who consume 100% fruit juice had a 16–19% lower frequency of anxiety than non-consumer adults. This is approximately a 20% lower number of days experiencing anxiety. While only about 1 day per month, this would be 12 days per year. Nutrition has been previously shown to play a role in prevention and treatment of anxiety [44]. A fruit- and vegetable-rich Mediterranean diet has been associated with a lower risk for anxiety disorders [45,46]. Fruit intake was associated with lower risk of anxiety among Iranian women [47] and a strict plant-based vegan diet has also been associated with lower anxiety [48] in population studies. A recent review of preclinical, observational, and experimental studies indicated a positive beneficial role of broad-spectrum micronutrient supplementation including vitamins B, C and E, zinc, magnesium and selenium, and a range of phytochemicals in anxiety [49]. However, we did not find any human epidemiological or clinical studies in our searches with PubMed and, to the best of our knowledge, our findings are the first human study reporting a protective association of 100% fruit juice consumption on anxiety.

There was a lower digital symbol score for 100% fruit juice consumers than non-consumers among adults age 60+ years. However, the differences were less than 1% and the scores for both consumers and non-consumers were well below the cutoff point (<80) for clinical disorder for dementia. We did not find any significant effects of 100% fruit juice intake on other measures of cognitive function in our regression analysis. There are only limited data on the effect of fruit juices on markers of cognitive functions. Regular consumption of fruit and vegetable juices was associated with a lower incidence of Alzheimer’s disease [50,51].

While Freije and colleagues [36] found that fruit juice consumption was not associated with poor mental health, they did not look specifically at measures of anxiety and depression. They also did not have a way to measure the amount of fruit juice consumed. We were able to obtain an estimate of the level of fruit juice consumption, and our results suggest that consumption of <12 ounces/day was associated with less anxiety, though this association was attenuated in a model including more covariates (model 3).

100% fruit juices are important source of key vitamins, minerals, and bioactive antioxidants such as flavonoids. In a recent cross-sectional analysis of NHANE data, 100% fruit juice made a significant contribution of the daily intakes of calcium (14%), magnesium (10%), potassium (16%), vitamin C (61%), and consumers of 100% fruit juice had significantly higher energy adjusted intakes of calcium (+8.0%), magnesium (+3.3%), potassium (+13.2%), thiamin (+5.1%), folate (+10.1%), vitamin B_6_ (+6.6%), vitamin C (+143%), Vitamin D (+17.8%), and beta-cryptoxanthin (+70.7%) than non-consumers [27]. Many of these vitamins, minerals, and flavonoids and other polyphenols have been hypothesized to have a beneficial role in cognitive health [44,51]. Higher intakes of antioxidants (vitamins C and E, and carotene) were reported to be associated with cognitive benefits in human studies [52,53,54]. Intake of orange juice rich in flavonoids was associated with the improvement of cognitive functions in acute clinical studies [55,56]. Supplementation with blueberry concentrate providing 387 mg anthocyanidins for 12 weeks also improved brain perfusion and activation in brain areas associated with cognitive function in healthy older adults [57]. Higher dietary intake of flavonoids (especially flavan-3-ols, catechins, and flavonols), and quercetin were associated with better cognitive health in an Italian cohort [58]. Flavonoids were recently reported to reduce neuroinflammation and oxidative stress, enhance cognitive function, attenuate cognitive decline, and reverse the symptoms associated with Alzheimer’s disease in experimental studies [59]. A recent review also indicated that phenolic acids that are present in some fruits may target multiple cellular pathways involved in the pathophysiology of cognitive disorders and thus may also exert neuroprotective effects [60].

Another key aspect of the present analysis is the use of a wide range of covariates including demographic, lifestyle, dietary, and health-related factors to adjust the results for potential confounding variables. In the present regression analysis, we also assessed the relationship of a number of possible covariables that may influence the relationship of 100% fruit juice intake with neurocognitive measures. The results obtained for Model 3 (Appendix A) showing that many of the additional variables added to the regression models were associated with neurocognitive measures could be useful for future research evaluating the association of neurocognitive measures with dietary intake and other characteristic of research subjects. Those pursing research in this area may want to consider including some or all of these variables in future research efforts.

A major strength of this study was the use of a large, nationally representative population-based sample of adults achieved through combining several sets of NHANES data releases and the use of key covariates to adjust data to remove potential confounding factors; however, even with these covariates, some residual confounders may still exist. An additional strength of this work is the assessment of a broad list of characteristics that might impact the relationship of fruit juice intake with neurocognitive measures. A major limitation of this study is the use of cross-sectional study design, which cannot be used to determine cause and effect. The use of self-reported 24 h dietary recalls relying on memory is prone to a potential source of bias for reporting of intake. While one of the best available and validated methodology, the AMPM method, was used to collect the dietary recalls in NHANES, there are still limitations with it [61]. Finally, future clinical studies are needed to better address whether 100% fruit juice consumption impacts neurocognitive function.

## 5. Conclusions

The results of this cross-sectional regression analysis show that 100% fruit juice consumption was not associated with many of the neurocognitive measures but was inversely associated with frequency of anxiety among adults and with lower digital symbol scores in those 60+ years. Additionally, a large number of potential confounding variables have been identified to be associated with one or more neurocognitive measures. These variables may be helpful to include in future studies evaluating the relationships of diet and neurocognitive measures.

## Figures and Tables

**Table 1 nutrients-14-04827-t001:** Neurocognitive markers measured in various NHANES cycles.

NHANES Cycles	Neurocognitive Markers Measured	Age Group (Years)
1988–1994	Simple Reaction Time Mean; Single Digit Learning Total Score; Symbol Digital Substitution Mean	20+
1999–2000	Anxious Month Period	20–39
2001–2002		
2003–2004		
2005–2006	Feeling Depressed Level	20+
2007–2008		
2009–2010		
2011–2012	Animal Fluency Score; CERAD: Score Delayed Recall; CERAD: Total Score Recall; Digital Symbol Score; Feeling Depressed Level; Trouble Remembering	60+ and 20+ (only for depression)
2013–2014		
2015–2016	Feel Anxious; Feeling Depressed Level	20+

**Table 2 nutrients-14-04827-t002:** Mean consumption and percentage of population consuming 100% juice.

Age (Years)	N	Day 1 Intake Data			Usual Intake (g/day)
		Consumers (N)	Consumers (%)	Intake (g)	
20+	62,606	14,172	20.8 ± 0.3	71.2 ± 1.5	69.1 ± 1.4
20–59	41,040	8051	18.4 ± 0.4	70.3 ± 1.8	67.7 ± 1.5
60+	21,566	6121	27.6 ± 0.6	73.8 ± 2.1	72.9 ± 1.9

Gender combined data, NHANES 1999–2018. Usual intakes were estimated by NCI method.

**Table 3 nutrients-14-04827-t003:** Association of usual intakes of 100% fruit juice with neurocognitive outcomes: consumer v. non-consumers and linear trend analyses. Gender combined data.

Age (Years)	Neurocognitive Outcome	Non-Consumer		Consumer		Linear Trend
		N	LSM ± SE	N	LSM ± SE	Beta
20+	Feel Anxious (days/month)	6325	6.14 ± 0.22	2043	5.14 ± 0.27 ^abc^	−0.005 ± 0.002 ^ab^
	Feeling Depressed Level (0–3)	20,590	0.33 ± 0.01	8705	0.32 ± 0.01	0.00002 ± 0.0001
20–59	Feel Anxious (days/month)	4197	6.79 ± 0.27	1225	5.53 ± 0.38 ^abc^	−0.007 ± 0.003 ^a^
	Feeling Depressed Level (0–3)	14,117	0.34 ± 0.01	5240	0.34 ± 0.01	0.0001 ± 0.0001
	Simple Reaction Time Mean (msec)	3450	234 ± 1	1013	231 ± 2	−0.012 ± 0.015
	Single Digit Learning Total Score (0–16)	3340	4.48 ± 0.12	986	4.06 ± 0.20	−0.002 ± 0.001
	Symbol Digital Substitution Mean (sec/digit)	3414	2.66 ± 0.02	1000	2.65 ± 0.03	0.00002 ± 0.0002
60+	Animal Fluency Score (0–40)	1696	17.9 ± 0.2	888	18.5 ± 0.3	0.001 ± 0.002
	CERAD: Score Delayed Recall (0–10)	1696	6.21 ± 0.10	899	6.23 ± 0.10	−0.001 ± 0.001 ^c^
	CERAD: Total Score Recall (0–30)	1697	19.6 ± 0.2	900	19.8 ± 0.2	0.001 ± 0.001
	Digital Symbol Score (0–133)	1658	52.4 ± 0.4	876	52.5 ± 0.5 ^c^	−0.006 ± 0.002 ^abc^
	Feel Anxious (days/month)	2128	4.40 ± 0.28	818	4.20 ± 0.46	0.001 ± 0.003
	Feeling Depressed Level (0–3)	6473	0.28 ± 0.01	3465	0.27 ± 0.01	−0.0001 ± 0.0001
	Trouble Remembering (0/1)	690	0.62 ± 0.03	388	0.67 ± 0.03	0.0005 ± 0.0003

Data adjusted for age, gender, ethnicity, physical activity level, poverty income ratio level, weight status, and current smoking status (Model 1). ^a^ significantly different from non-consumer or significant linear trend at *p* < 0.05. ^b^ significantly different from non-consumer or significant linear trend at *p* < 0.05 after adjusting the data for Model 1 covariates plus Healthy Eating Index 2015 (Model 2). ^c^ significantly different from non-consumer or significant linear trend at *p* < 0.05 after adjusting the data for Model 2 covariates plus weekday hours of sleep, antidepressant medication use, hypertension medication use, education level, glycohemoglobin, elevated blood pressure, and doctor told you had a stroke, and intakes of caffeine, vitamin B_12_, iron, folic acid, total fruits, whole fruits, and total vegetables excluding legumes (Model 3). See Table 1 for NHANES cycles used for each neurocognitive marker.

**Table 4 nutrients-14-04827-t004:** Association of usual intakes of 100% fruit juice by consumption levels and neurocognitive outcomes. Gender combined data.

Age (Years)	Neurocognitive Outcome	100% Fruit Juice Consumption Levels					
		0 oz	>0 to 4 oz	>4 to 8 oz	>8 to 12 oz	>12 oz	Group Trend
20+	Feel Anxious (days/month)	6.14 ± 0.22	4.88 ± 0.44 ^abc^	5.39 ± 0.52	4.87 ± 0.57 ^a^	6.58 ± 2.80	−0.436 ± 0.161 ^ab^
	Feeling Depressed Level (0–3)	0.33 ± 0.01	0.32 ± 0.01	0.31 ± 0.01	0.29 ± 0.02	0.37 ± 0.11	−0.006 ± 0.007
20–59	Feel Anxious (days/month)	6.79 ± 0.27	5.46 ± 0.62 ^ab^	5.52 ± 0.65	5.51 ± 0.67	7.38 ± 3.01	−0.563 ± 0.198 ^ab^
	Feeling Depressed Level (0–3)	0.34 ± 0.01	0.34 ± 0.02	0.35 ± 0.02	0.31 ± 0.03	0.40 ± 0.13	−0.0001 ± 0.008
	Simple Reaction Time Mean (msec)	234 ± 1	233 ± 5	231 ± 2	228 ± 3	235 ± 16	−1.239 ± 1.065
	Single Digit Learning Total Score (0–16)	4.48 ± 0.12	4.16 ± 0.35	4.04 ± 0.26	4.08 ± 0.47	3.72 ± 0.54	−0.196 ± 0.099
	Symbol Digital Substitution Mean (sec/digit)	2.66 ± 0.02	2.69 ± 0.07	2.63 ± 0.04	2.65 ± 0.07	2.70 ± 0.13	−0.009 ± 0.017
60+	Animal Fluency Score (0–40)	17.9 ± 0.2	18.6 ± 0.4	18.5 ± 0.4	18.1 ± 0.9	19.1 ± 1.5	0.254 ± 0.184
	CERAD: Score Delayed Recall (0–10)	6.21 ± 0.10	6.31 ± 0.14	6.28 ± 0.12	5.73 ± 0.40	5.70 ± 0.87	−0.034 ± 0.058
	CERAD: Total Score Recall (0–30)	19.6 ± 0.2	20.0 ± 0.2	19.7 ± 0.3	18.6 ± 0.8	20.4 ± 0.9	0.037 ± 0.108
	Digital Symbol Score (0–133)	52.4 ± 0.4	53.7 ± 1.1	52.1 ± 0.8 ^c^	51.2 ± 1.7	46.8 ± 2.3 ^abc^	−0.338 ± 0.263 ^c^
	Feel Anxious (days/month)	4.39 ± 0.28	3.49 ± 0.71	5.01 ± 0.68	4.07 ± 1.38	2.55 ± 2.25 ^c^	0.041 ± 0.282
	Feeling Depressed Level (0–3)	0.28 ± 0.01	0.28 ± 0.02	0.25 ± 0.02	0.29 ± 0.04	0.25 ± 0.07	−0.011 ± 0.009
	Trouble Remembering (0/1)	0.62 ± 0.03	0.60 ± 0.05	0.72 ± 0.04	0.79 ± 0.10 ^c^	0.64 ± 0.19	0.043 ± 0.021 ^a^

Data presented as least square mean (LSM) ± standard error (SE). Data adjusted for age, gender, ethnicity, physical activity level, poverty income ratio level, weight status, and current smoking status (Model 1). ^a^ significantly different from 0 oz or significant group trend at *p* < 0.05. ^b^ significantly different from non-consumer or significant linear trend at *p* < 0.05 after adjusting the data for Model 1 covariates plus Healthy Eating Index 2015 (Model 2). ^c^ significantly different from non-consumer or significant linear trend at *p* < 0.05 after adjusting the data for Model 2 covariates plus weekday hours of sleep, antidepressant medication use, hypertension medication use, education level, glycohemoglobin, elevated blood pressure, and doctor told you had a stroke, and intakes of caffeine, vitamin B_12_, iron, folic acid, total fruits, whole fruits, and total vegetables excluding legumes (Model 3). See Table 1 for NHANES cycles used for each neurocognitive marker.

## Data Availability

All data obtained for this study are publicly available at: http://www.cdc.gov/nchs/nhanes/ (4 August 2021).

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
