# Peer review of "Association of 100% Fruit Juice Consumption with Cognitive Measures, Anxiety, and Depression in US Adults"

_nutrients, 2022, doi:10.3390/nu14224827_

Round 1
Reviewer 1 Report
This meta-analysis is very interesting, however, some curiosity need to be cleared
- The grouping is confusing, is that '>' or '≥'. Use of > only conflicts with the next group. Please confirm and justify.
- Correction or possibilities of biasedness is not cleared, e.g. amount of juice. Large volume of juice may have impact on social behavior. Also, the quality and fruit specificity could be a cognitive note.
- Forgetting keys or wallet could be a sign of work or social stress. Author could have asked more specific question that regard work memory or spatial memory.
- In my opinion, all the preset questions are not specific for cognitive evaluation, but they are depression like or mental health related questions. Questions like forgetting name or location of a large things like car should have been included.
- Is that possible that any fruit juice can modulate cognitive function? Was author considering 100% fruit juice consume changes mood and that helps cognition? The title and complete correlation of findings are confusing. Could be because of no biochemical presentation.
Author Response
Reviewer 1 comments
This meta-analysis is very interesting, however, some curiosity need to be cleared
- The grouping is confusing, is that '>' or '≥'. Use of > only conflicts with the next group. Please confirm and justify.
- Correction or possibilities of biasedness is not cleared, e.g. amount of juice. Large volume of juice may have impact on social behavior. Also, the quality and fruit specificity could be a cognitive note.
- Forgetting keys or wallet could be a sign of work or social stress. Author could have asked more specific question that regard work memory or spatial memory.
- In my opinion, all the preset questions are not specific for cognitive evaluation, but they are depression like or mental health related questions. Questions like forgetting name or location of a large things like car should have been included.
- Is that possible that any fruit juice can modulate cognitive function? Was author considering 100% fruit juice consume changes mood and that helps cognition? The title and complete correlation of findings are confusing. Could be because of no biochemical presentation.
Authors Response: Thank you for the comments/suggestion above. Deleted '≥' to avoid confusion. The grouping now includes “>0 to 4, >4 to 8, >8 to 12, and >12”.
Unfortunately, we did not have the luxury of selecting questions for this study. Our data comes from the large, publically available NHANES data sets. Having more practical measures of memory/spatial memory would have been better, but we can only analyze the variables we have in this data set. Also, while there are numerous biochemical variables in the NHANES data set, it is unclear if any measure relates to cognitive function. Note, we did assess whether certain laboratory values (i.e., glycohemoglobin, and blood levels of vitamin B12, iron, and folate) were associated with cognitive, anxiety, and depression variables (see supplemental table 1).
Reviewer 2 Report
In the manuscript entitled as “Association of 100% fruit juice consumption with cognitive measures, anxiety, and depression in US adults”, Sanjiv Agarwal et al. tried to evaluate whether 100% fruit juice consumption has an association with cognition, anxiety, and depression. To this end, authors used public database being constructed through NHANES by NCHS. The topic was interesting, and the method of this study was proper to excludes bias. As a result, authors conclude that consumers of 100% fruit juice had 16.3% fewer days per month of feeling anxious compared to non-consumers. Because it was made from analysis of 62,606 adults.
Author Response
Reviewer 2 comments
In the manuscript entitled as “Association of 100% fruit juice consumption with cognitive measures, anxiety, and depression in US adults”, Sanjiv Agarwal et al. tried to evaluate whether 100% fruit juice consumption has an association with cognition, anxiety, and depression. To this end, authors used public database being constructed through NHANES by NCHS. The topic was interesting, and the method of this study was proper to excludes bias. As a result, authors conclude that consumers of 100% fruit juice had 16.3% fewer days per month of feeling anxious compared to non-consumers. Because it was made from analysis of 62,606 adults.
Authors Response: Thank you for these comments (no changes made to the manuscript based on these comments)
Reviewer 3 Report
I would encourage resubmission after adding the latest data of the years 2019-2022. Furthermore, there are several grammatical mistakes in this manuscript for example in Abstract Line 16 Regression analyses. The English language should be improved by a native speaker. In key words National Health and Nutrition Examination Survey and again the acronyms have been used as NHANES. Before submitting the manuscript again, the authors should change the sentence line 70-72 as well. The authors should check the following manuscript entitled “Association Between Consumption of Sugar Sweetened Beverages and 100% Fruit Juice with Poor Mental Health Among US Adults in 11 US States and the District of Columbia” which has been published in May 2021 and another one on July 15th 2022. Please also check this article published in Nutrients in May 2022
https://doi.org/10.3390/nu14102127
The authors should know that they are targeting MDPI Journals and then Q1 Journal Nutrients.
I have shared some recently published articles doi, the authors should improve their manuscript by reading this manuscript and must include the latest data.
http://dx.doi.org/10.5888/pcd18.200574
https://doi.org/10.1186/s12905-022-01870-3.
https://doi.org/10.3389/fpsyg.2022.1024946
https://doi.org/10.3390/nu14102127
Author Response
Reviewer 3 comments
I would encourage resubmission after adding the latest data of the years 2019-2022. Furthermore, there are several grammatical mistakes in this manuscript for example in Abstract Line 16 Regression analyses. The English language should be improved by a native speaker. In key words National Health and Nutrition Examination Survey and again the acronyms have been used as NHANES. Before submitting the manuscript again, the authors should change the sentence line 70-72 as well. The authors should check the following manuscript entitled “Association Between Consumption of Sugar Sweetened Beverages and 100% Fruit Juice with Poor Mental Health Among US Adults in 11 US States and the District of Columbia” which has been published in May 2021 and another one on July 15th 2022. Please also check this article published in Nutrients in May 2022 https://doi.org/10.3390/nu14102127
The authors should know that they are targeting MDPI Journals and then Q1 Journal Nutrients.
I have shared some recently published articles doi, the authors should improve their manuscript by reading this manuscript and must include the latest data.
http://dx.doi.org/10.5888/pcd18.200574
https://doi.org/10.1186/s12905-022-01870-3.
https://doi.org/10.3389/fpsyg.2022.1024946
https://doi.org/10.3390/nu14102127
Authors Response: While we also would like to include the most up-to-date NHANES data, unfortunately, given COVID, the 2019-2022 data are not complete (there are no dietary data available to assess intake of fruit juice).
Abstract line 16 modified and lines 70-72 are revised; we reviewed the manuscript again and did not see the need to make any additional grammatical changes.
We have reviewed all of the articles suggested by this reviewer Out of 4 references only one related to fruit juice and mental health. We have included a description of findings of this publication in the introduction and then related the findings of this publication to our findings in the discussion.
One paper was on fruit juice and mortality (including in particular cardiovascular mortality), another was on sugar sweetened beverages (where the authors included fruit juices as a sugar sweetened beverage when in the US 100% fruit juice is not considered a sugar sweetened beverage) and the last paper was on Healthy Beverage Index which included all types of beverages and not just 100% fruit juice. Given these articles are not within the scope of our manuscript, these were not included in our revised manuscript.
Round 2
Reviewer 1 Report
Appreciate author's response. Perhaps addressing as limitations of this study to properly evaluate cognitive function would be better. Because all these questions and observations mostly related to stress. I suggest add limitations in either discussion or conclusion section.
Author Response
Reviewer comments
Appreciate author's response. Perhaps addressing as limitations of this study to properly evaluate cognitive function would be better. Because all these questions and observations mostly related to stress. I suggest add limitations in either discussion or conclusion section.
Authors Response: Thank you for the comments/suggestion. We have added a statement at the end of discussion in the limitations. We have also done a spell check and made minor corrections throughout the manuscript.
Reviewer 3 Report
The authors have improved the manuscript. It can be accepted now.
Author Response
Reviewer comments
The authors have improved the manuscript. It can be accepted now.
Authors Response: Thank you for these comments. We have done a spell check and made minor corrections throughout the manuscript.